# Podoplanin Expression Independently and Jointly with Oral Epithelial Dysplasia Grade Acts as a Potential Biomarker of Malignant Transformation in Oral Leukoplakia

**DOI:** 10.3390/biom12050606

**Published:** 2022-04-19

**Authors:** Luís Monteiro, Barbas do Amaral, Leonor Delgado, Fernanda Garcês, Filomena Salazar, José Júlio Pacheco, Carlos Lopes, Saman Warnakulasuriya

**Affiliations:** 1Medicine and Oral Surgery Department, University Institute of Health Sciences (IUCS), Cooperativa de Ensino Superior Politécnico e Universitário (CESPU), 4585-116 Gandra, Portugal; barbas.jm.amaral@gmail.com (B.d.A.); filomena.salazar@cespu.pt (F.S.); julio.pacheco@iucs.cespu.pt (J.J.P.); 2UNIPRO, Oral Pathology and Rehabilitation Research Unit, University Institute of Health Sciences (IUCS), Cooperativa de Ensino Superior Politécnico e Universitário (CESPU), 4585-116 Gandra, Portugal; mldelgado.mv@gmail.com (L.D.); fernanda.garcez@cespu.pt (F.G.); carlos.lopes@iucs.cespu.pt (C.L.); 3Stomatology Department, Hospital de Santo António, Centro Hospitalar do Porto, 4099-001 Porto, Portugal; 4Molecular Pathology and Immunology Department, Institute of Biomedical Sciences Abel Salazar (ICBAS), Porto University, 4099-001 Porto, Portugal; 5Faculty of Dentistry, Oral & Craniofacial Sciences, King’s College London, The WHO Collaborating Centre for Oral Cancer, London SE1 9RT, UK; saman.warne@kcl.ac.uk

**Keywords:** oral leukoplakia, podoplanin, CD44v6, CD147, EGFR, p53, p16, dysplasia, malignant transformation

## Abstract

Our aim was to evaluate the expression of biomarkers, CD44v6, CD147, EGFR, p53, p63, p73, p16, and podoplanin in oral leukoplakias (OL) and to assess their potential for prediction of malignant transformation (MT). We analyzed the expression of CD44v6, CD147, EGFR, p53, p63, p73, p16, and podoplanin by immunohistochemistry in 52 OL, comprised of 41 low-grade (LG) dysplasia and 11 high-grade (HG) cases. Twelve healthy normal tissues (NT) were also included. Univariate and multivariate analysis were performed to evaluate any association with MT. Variable expression among the studied markers was observed, with a significant increase of high expression from NT to LG and HG cases in CD44v6 (*p* = 0.002), P53 (*p* = 0.002), P73 (*p* = 0.043), and podoplanin (*p* < 0.001). In multivariate analysis, cases with high podoplanin score showed a significant increased risk of MT (HR of 10.148 (95% CI of 1.503–68.532; *p* = 0.017). Furthermore, podoplanin combined with binary dysplasia grade obtained a HR of 10.238 (95% CI of 2.06–50.889; *p* = 0.004). To conclude, CD44v6, p53, p73, and podoplanin showed an increasing expression along the natural history of oral carcinogenesis. Podoplanin expression independently or combined with dysplasia grade could be useful predictive markers of MT in OL.

## 1. Introduction

Oral potentially malignant disorders (OPMD), were defined by the World Health Organization (WHO) Collaborating Centre for Oral Cancer in 2020, as lesions or conditions that carry a significant risk for malignant transformation and include oral leukoplakia (OL), proliferative verrucous leukoplakia (PVL), oral erythroplakia (OE), oral lichen planus (OLP), oral lichenoid lesions (OLL), chronic graft-versus-host disease (cGVHD), oral submucous fibrosis (OSMF), and actinic cheilitis [1]. OL, defined as “a white plaque with a questionable risk of cancer that can only be diagnosed once other specific conditions have been ruled out” [2], is one of the most common OPMD with a worldwide prevalence reported as 2.6 to 4.1% with possible geographic variations [3,4]. The malignant transformation (MT) rates of OL have been reported as 9.8% (95% CI: 7.9–11.7) by recent systematic reviews [5]. OL can precede oral cancer development and therefore their detection and management could contribute to a reduction in cancer incidence [2,6,7].

Nevertheless, establishing the risk of malignant transformation could be challenging [8]. Some histopathological characteristics, such as the dysplasia grade have demonstrated some prognostic value [6,9,10], but grading of oral epithelial dysplasia could be subjective, which limits its value in risk assessment [11,12]. Taking in to account the importance of risk stratification of patients with OL to plan treatment, there is a need for better predictive biomarkers of transformation.

Oral carcinogenesis involves multiple molecular or genetic alterations in cellular pathways providing the acquisition of some capabilities such as continuous and autonomous proliferative cell phenotype, invasion and neoangiogenesis, reprograming cellular energetics, or evading immune destruction that transform a single cell or a clone of cells into a malignant tumor [13,14,15,16,17,18]. Many of these deregulated genes and proteins could be indicators of an active ongoing carcinogenesis process and could act as potential predictive biomarkers in OPMD [17,19,20,21]. We have previously reported the abnormal expression of several proteins in oral squamous cell carcinomas, such as EGFR, p53, p16, podoplanin, or CD147 [22,23,24], and we hypothesized “could these proteins be abnormally expressed in OL?”. Any findings could then be translated to predict the risk of malignant transformation of OL. Examining biomarkers in different stages of progression from OPMD to oral squamous cell carcinoma has been reported [25] and may allow for the characterization of genes and proteins expressed during the process of carcinogenesis. The benefits of using combined factors to investigate OPMDs instead of applying a single biomarker was remarked on at the inaugural Global Oral Cancer Forum [26] and more research should address the role of these biomarkers in the prognostification of OPMDs.

The aim of this study was to evaluate the expression of biomarkers, CD44v6, CD147, EGFR, p53, p63, p73, p16, and podoplanin in OL, and specifically to identity which of these biomarkers could be associated with the risk of malignant transformation of this disorder.

## 2. Materials and Methods

### 2.1. Population

We performed an observational retrospective cohort study on OL submitted to histopathological evaluation in the pathology service of the *Hospital de Santo António* (HSA), in the Centro Hospital do Porto, Portugal, during the period of 1995 to 2006. The study was approved by the Institutional Ethical Board of the hospital (DEFI; 024/CES/03) and performed in accordance with the Declaration of Helsinki. We reviewed and extracted clinical and pathological information from the pathology department and clinical hospital databases including patients’ age, gender, lesion location, clinical type of the lesions and tabulated them according to Warnakulasuryia et al. [1], tobacco and alcohol habits, primary treatment, and follow-up information of malignant transformation of previously diagnosed OL.

A consecutive cohort of patients was included that had clinical diagnosis of OL located in the oral cavity (ICD 10: C00-06) confirmed by histopathology, including either epithelial hyperplasia or hyperkeratosis/or parakeratosis without dysplasia (sometimes designated as “keratosis of unknown significance”—KUS) [27,28] or with epithelial dysplasia. Additionally, we included 12 cases of healthy normal oral mucosa from the same period of study. We excluded patients younger than 18 years old, patients that had received previous treatments for oral cancer such as radiotherapy or chemotherapy, or cases without pathological confirmation of diagnosis. Histopathological analysis was performed on new 3 µm sections stained with haematoxylin-eosin for all cases. Dysplasia was graded by two experienced observers using the binary dysplasia grade system [29] into the two groups of low grade and high grade. In this classification, the cut-off point between “high-risk” vs. “low-risk” lesions was made considering at least four architectural changes and five cytological changes for “high-risk” lesions (including the architecture criteria: 1—irregular epithelial stratification, 2—loss of polarity of basal cells, 3—drop-shaped rete ridges, 4—increased number of mitotic figures, 5—abnormally superficial mitoses, 6—premature keratinization in single cells, 7—keratin pearls within rete ridges; and cytology criteria: 1—abnormal variation in nuclear size, 2—abnormal variation in nuclear shape, 3—abnormal variation in cell size, 4—abnormal variation in cell shape, 5—increased nuclear– cytoplasmic ratio, 6—increased nuclear size, 7—atypical mitotic figures, 8—increased number and size of nucleoli, 9—hyperchromatism). If there was discrepancy in reporting by the two observers, the grade was agreed by consensus. In the case of multiple lesions or multiple samples found on the same case, we selected the most representative and more advanced lesion for grading and biomarker evaluation.

### 2.2. Immunohistochemistry

The 3 μm sections were deparaffinized and epitope retrieval was performed by immersing in a water bath at 98 °C for 30 min, with 0.01 M citrate buffer (pH 6.0) for anti-podoplanin, anti-CD44v6, and anti-CD147, and with EDTA buffer 0.01 M (pH 9.0) for anti-p63 and anti-EGFR. After blocking for non-specific binding, the slides were incubated with primary monoclonal antibodies (Table 1).

The immunodetection was performed using a peroxidase-labelled indirect polymer (NovoLinkTM Polymer Detection System; Novocastra, Leica Biosystems Newcastle Ltd., Newcastle upon Tyne, UK.) revealed with DAB chromogen (diaminobenzidine). Finally, the slides were counterstained with Gill´s Hematoxylin and mounted with a hydrophobic medium. In each staining run, we used both positive (breast carcinoma, normal skin, oral mucosa, and tonsil) and negative (omission of primary antibody) controls.

### 2.3. Evaluation of Immunohistochemical Expression

All sections were evaluated by two authors blinded to clinical variables, using a ZEISS AxioLab A1^®^ microscope (Carl Zeiss Microscopy GmbH, Jena, Germany), with a ZEISS Axiocam 105 color^®^ and ZEISS Zen2^®^ software. In the presence of a discordant result between the two authors’ evaluation, a review of the slides was performed under a multi-head microscope to achieve a consensus.

Considering the different types of included lesions, we performed a combination of extent and intensity of staining into a combined score, using a similar cut-off for low and high expression for all markers. For this, initially, we classified the percentage of stained cells in: [0 (negative, or <10%); 1 (10–24%); 2 (25–49%); 3 (50–74%); and 4 (>75%)] and intensity in [0 (absent), 1 (weak), 2 (moderate), and 3 (strong)] [23]. Then, we combined intensity and percentage of positive cells to construct the final marker score, from 0 to 7 [(0–1) negative; (2–3) I score; (4–5) II scotore; (6–7) III score]. For data analysis, scores 0/I were considered as low expression and scores II/III as high expression of individual markers, as previously reported [23].

### 2.4. Statistical Analysis

The results are presented in relative and absolute frequencies. The associations between categorical variables were evaluated by Kruskal–Wallis one-way ANOVA test with pairwise multiple comparisons (with bonferroni adjustment when applicable). The correlation between biomarkers was performed using Spearman’s correlation coefficient test. Oral malignant transformation-free survival (MTFS) rate was evaluated in all OL cases and was defined as the time interval (months) between diagnosis and the first histological confirmed oral cancer. The Kaplan–Meier method with log-rank test was used to evaluate the predictive effect of variables on MTFS in a univariate analysis. Then, the Cox proportional hazards model was used to investigate the independent value of the variables with significance in the univariate analyses. The level of significance used for the statistical tests was 5%. Statistical analysis was performed using IBM SPSS Statistics version 24.0 software (IBM Corporation, Armonk, NY, USA).

## 3. Results

The final sample composed of 64 patients, 46 (71.9%) males and 18 (28.1%) females with a mean age of 58.1 ± 16.8 years, including 52 (81.2%) OL and 12 (18.8%) cases of healthy oral normal tissues (NT). Demographic and clinical features were already described previously [30]. Briefly, most of the cases were located on the tongue (27; 42.2%), followed by 13 cases on the buccal mucosa (20.3%), 11 (17.2%) on the alveolar bridge/gingiva mucosa, 6 (9.4%) on the labial mucosa, 4 (6.3%) on the retromolar trigone, 2 cases (3.1%) on the floor of the mouth, and one (1.6%) case on the hard palate. There were 39 (90.7%) cases with a single lesion and 4 (9%) with multiple lesions (information available in 43 patients). OL cases were classified (according to WHO classification) [2] under homogeneous type in 17 (63%) patients and non-homogeneous type in 10 (27%) patients where this information was available. Using the binary histological grade classification of tissue dysplasia, there were 41 (78.8%) low-grade and 11 (21.2%) high-risk grade OL. Regarding the type of management performed, in 36 patients (69.2%) OL were excised with scalpel, 5 (9.6%) submitted to laser ablation, and 11 (21.2%) cases to non-surgical management. Seventeen (40.5%) patients with OL were current smokers, and 15 (56.2%) were current alcohol beverage consumers.

### 3.1. Protein Analysis

#### 3.1.1. CD44v6

The expression of CD44v6 was observed as membranous in all analyzed available specimens (*n* = 59), from a weak expression on basal layers to a more diffuse pattern on multiple layers of low- and high-dysplasia-grade cases (Figure 1). CD44v6 high expression was observed in seven cases (58.3%) of normal tissues, 35 (94.6%) of low-grade lesions and ten (100%) of high-grade lesions (*p* = 0.002, two-side overall test). Significant differences were obtained specially between NT and LG lesions (*p* = 0.002) and between NT and HG lesions (*p* = 0.009). No association was found between CD44v6 expression and other clinical-pathological variables.

#### 3.1.2. CD147

Most cases showed membranous expression of CD147 more restricted to the basal layer, except five cases where no staining was found, one in normal tissue, and four in low-grade lesions, in the 56 cases available for this marker (Figure 1). High expression of CD147 was observed in 6 cases (50%) of NT, 17 (47.2%) on LG, and 7 (87.5%) on HG lesions (*p* = 0.118, two-side overall test). No association was found between CD147 expression and other clinical-pathological variables.

#### 3.1.3. EGFR

All cases presented some level of EGFR membranous expression, often with a continuous and diffused expression pattern over the epithelial layers (Figure 1). High EGFR expression was observed in 11 cases (91.7%) of normal tissues, 34 (94.4%) in LG, and 9 (100%) in HG lesions, in the 58 cases available for this marker (*p* = 0.698). No association was found between EGFR expression and other clinical-pathological variables.

#### 3.1.4. Podoplanin

Podoplanin was detected at the cell membrane of many of the included cases except in 3 NT cases (25%), 13 (50%) low-grade lesions, and 3 cases (23%) of high-grade lesions. Podoplanin was also detected in the lymphatic vessels in every case (serving as positive internal control) (Figure 1). The expression patterns ranged from focal cellular points on the basal layer to a more continuous staining along basal layers or extending focally to more upper layers. High podoplanin expression cases were only observed in one case (3.3%) of the LG group and in four (44.4%) of the HG group in the 51 cases available for this marker (*p* < 0.001, two-side overall test). Significant differences were obtained specially between NT and HG (*p* = 0.002) and between LG and HG (*p* = 0.001).

Higher expression of podoplanin was significantly associated with cases with multiple lesions (*p* = 0.009). We also performed the analysis using the score according Kawaguchi et al. [31], observing a high score in one (3.3%) case of NT, nine (30%) of LG cases, and seven (87.5%) HG cases (<0.001). Again, significant differences were obtained, especially between NT and HG (*p* = 0.001) and between LG and HG (*p* = 0.008). 

Podoplanin’s higher score was significantly associated in cases with multiple lesions (*p* = 0.033).

#### 3.1.5. P53

Six cases (11.7%) were completely negative for nuclear p53 expression, three in normal tissues, and three in low-grade lesions (Figure 2). High expression of p53 was observed in 1 case (9.1%) of normal tissues (restricted to basal and suprabasal layers), 20 (60.6%) in LG lesions, and 5 (83.3%) in HG lesions, in the 51 cases available for this marker (*p* = 0.002, two-side overall test). Significant differences were obtained in particular between NT and LG (*p* = 0.006) and NT and HG (*p* = 0.009). No association was found between p53 expression and other clinical-pathological variables.

#### 3.1.6. P63

All cases showed some degree of p63 nuclear expression in 55 available cases, many times extending up to more than half of the thickness of the epithelia (Figure 2). High p63 expression was observed in 12 cases (100%) of NT, 34 (97.1%) cases in LG, and 8 (100%) in HG cases (*p* = 0.751). No association was found between p63 expression and other clinical-pathological variables.

#### 3.1.7. P73

Nuclear expression of p73 protein was not observed in two normal tissue cases, four cases of LG and HG lesions (Figure 2). High expression of p73 corresponded to four cases (33.3%) of normal tissues, 24 (72.7%) of LG, and 6 (75%) of HG, in the 53 cases available for this marker (*p* = 0.043, two-side overall test). Significant differences were obtained especially between NT and LG cases (*p* = 0.047). 

p73 higher expression was related with age (>58 years-old) (*p* = 0.014) and with location of the lesion (*p* = 0.04).

#### 3.1.8. P16

Both nuclear and/or cytoplasmic p16 expression was observed in 44.2% (*n* = 23) of the available cases (*n* = 52), being absent in 4 normal tissue cases, 22 in low-grade lesions, and 3 cases in high-grade lesions (Figure 2). High expression of p16 was observed in two cases (16.7%) of normal tissues, four (12.9%) of LG, and four (44.4%) of HG (*p* = 0.108). Using the eighth AJCC p16 cut-off [32], we detected only two cases (3.8%) with positive p16AJCC score (one in each LH and HG groups) and without statistical association with the groups of NT, LG, and HG lesions (*p* = 0.414).

No relation was found between p16 expression and other clinical-pathological variables.

### 3.2. Correlation between Markers

A Spearman correlation was found only between CD44v6 and EGFR (rho of 0.390; *p* = 0.003) and between p53 and p73 (rho of 0.439; *p* = 0.02).

### 3.3. Analysis of Oral Malignant Transformation Free Survival

We performed an analysis of the oral malignant transformation rate among the OL cases with a mean follow-up period of 32.4 ± 29 months (minimum 2 months; maximum 120 months). We observed six (11.5%) cases of malignant transformation included in this analysis resulting in a cumulative five-year malignant transformation-free survival (MTFS) rate of 83.2%.

On univariate analysis of MTFS, the only clinical-pathological variable associated with MTFS was the binary dysplasia grade (*p* < 0.001). Podoplanin expression score was the only biomarker with a significant relation with MTFS (*p* < 0.001) (Table 2).

In the multivariate analysis, we observed that podoplanin score (with combination of extent and intensity) presented an independent association on the risk of transformation to oral cancer (*p* = 0.017; HR of 10.148, 95% CI of 1.503–68.532), while binary dysplasia grade reached marginal significance (*p* = 0.054; HR of 9.316, 95% CI of 0.965–89.951).

Interestingly, when we included in multivariate analysis the podoplanin score (with extent and intensity combination) and score used by Kawaguchi et al. [31], we observed that only the first presented a significant effect (*p* = 0.022, HR of 13.009, 1.439–117.601), while the last lost significance (*p* = 0.697). Also, if we include Kawaguchi score with binary dysplasia grade, we observed that only binary grade classification presented a significant and independent association with MTFS in our sample (*p* = 0.03, HR of 13.358, 95% CI of 1.286–138.801). 

These results allow us to evaluate a combined score between podoplanin (extent and intensity) score and binary dysplasia grade, where both high podoplanin expression and high dysplasia grade were considered as high-risk vs. low-risk for other combinations, as can be observed in Figure 3 (using the Kaplan–Meier estimates of probability of survival).

Using the Cox regression method, we observed a HR of 10.238 (95% CI of 2.06–50.889; *p* = 0.004) for combined dysplasia grade and podoplanin high-score cases compared with cases of low dysplasia grade and with low podoplanin expression.

## 4. Discussion

The assessment of potential risk factors for malignant transformation of OPMDs has been a challenging task with conflicting results reported in the scientific literature [8,16,17,19,21,33,34]. The last World Workshop on Oral Medicine VII reviews on biomarkers in OL reported that there was no evidence to support translation of candidate biomarkers predictive of malignant transformation of OL [33,35]. The latest review on biomarkers from the WHO Collaborative Centre for Oral Cancer 2021 Workshop on OL revealed 46 eligible studies on this topic with 49 biomarkers studied in OL [21]. We developed this study based on the evidence in this latest SR, which identified at least one study on each of the respective biomarkers selected by us for this investigation. The objective therefore was to analyze the expression of several proteins, including CD44v6, CD147, EGFR, p53, p63, p73, p16, and podoplanin, in a single cohort of OL and to study their potential value for predicting the probable malignant transformation in patients with OL.

We observed an increasing expression of almost all the markers examined from normal oral tissues to low-grade and high-grade cases. This was more evident for CD44v6, P53, p73, and podoplanin. Although in our study we report the increased expression of these proteins from NT high grades of dysplasia, it is difficult to define if these molecules are drivers or a consequence of the oral carcinogenesis process.

CD44v6, an adhesion molecule from the CD44 transmembrane glycoprotein family, has been particularly involved during evolution of several cancers and incorporates some reported possible stem cell properties [36,37,38]. Although we note a diffuse and continuum expression over the epithelium, especially in the proliferation compartment, an increase of expression score was observed from NT to LG or HG, which could suggest a possible contribution of this marker to oral carcinogenesis. However, reported studies did not find any significant alteration of this biomarker during oral carcinogenesis [36,39]. A decrease in CD44v6 expression on oral dysplasia was reported by Godge and Poonja [40]. In fact, conflicting results have been reported for this biomarker with overexpression or subexpression in potentially malignant disorders and malignant oral tumors [23]. Podoplanin, the other biomarker with stem cell properties, was firstly reported on kidney podocytes, and is considered a marker of lymphatic endothelium [41]. The focal and cluster expression of podoplanin in our samples also suggests a stem cell function of this protein, as reported by some authors [42,43,44], increasing during carcinogenesis, and acting as a clonal proliferation boost for malignant transformation of OL [45,46]. An association of podoplanin with oral dysplasia was reported by other authors [43,46,47,48,49,50,51]. The use of other cancer stem cells markers was suggested as prognostic markers for malignant progression of OPMDs [52]. P53 is one of the most studied proteins in cancer, and has been associated with oral carcinogenesis and prognosis [53,54,55,56]. We observed differences in p53 expression among the group of lesions evaluated. The deregulation of this oncosupressor protein could be important for the acquisition of additional errors or tumor capacities on carcinogenesis. In fact, Sunberg et al. [57] showed a significant relation of p53 with recurrence of OL. P73 was also increased in our cases series. The role of P73 in oral carcinogenesis is less clear and poorly studied than the other P53 protein family members, however [58].

One of the main objectives of this paper was to determine the translational value of these proteins as biomarkers of oral malignant transformation. In our cohort of patients, podoplanin was the only protein that was related with MT and in the multivariate analysis was confirmed to be an independent and significant biomarker. Such an association using multivariate analysis was reported in a few earlier studies on OL [31,43,50]. Kawaguchi et al. [31] reported an association of podoplanin with MT in OL, using a score based on the expression of clusters within the epithelium. In previous research on podoplanin, we noted that in some cases, the intensity of podoplanin could differ from the control tissue such as lymphatic vessels. We decided to use a combined score of extension and intensity for every biomarker analyzed in this study, promoting the same conditions for all markers. Podoplanin has been related with several capabilities that enhance tumoral properties such as cell proliferation, cell motility, cell loss-attachment, and invasion by stimulating metalloproteinases such as MMP-9 or downregulating E-cadherin [23,43,44,59,60]. We observed in the present sample of cases that combining the extent and intensity characteristics could give more prognostic information than other score combinations in the present sample.

We could not find an association of MT with other biomarkers examined in our study. A few of these biomarkers have been evaluated in longitudinal studies with multivariate analysis and described as independent biomarkers in OL [44,61,62,63,64,65,66,67,68,69]. These differences in the results could be related to the different sample composition of populations and methodologies used in the different studies. This points out the need for the development of multicenter studies using the same diagnostic criteria and similar methodologies. 

A recent systematic review found evidence that moderate/severe dysplasia grades were predictive of malignant transformation [70]. An expert group also reported the importance of this dysplasia grade in OL [71]. We observed a marginal significant association of binary dysplasia grade with the malignant transformation when podoplanin score was also in multivariate analysis. Epithelial dysplasia grade is currently one of the most frequently used tools for the risk assessment of OL in clinical practice and the binary reporting of dysplasia grade has been recommended for future use [72,73]; moreover, its predictive capacity for the malignant transformation of OL has been supported in past studies [6,9]. However, there are studies reporting that relying only on dysplasia grading could be insufficient and subjective in clinical use [11,12]. Based on the available evidence, podoplanin offers promise as a useful biomarker for further clinical research [68,74]. Biomarkers associated with malignant transformation of OL, independently of dysplasia grade, could facilitate the objective assessment of the malignant potential of these disorders.

We acknowledge some limitations of the work reported here, such as the retrospective nature of the study, the small sample size, potential variations related with immunohistochemistry technique and evaluation, and our report being a single-institution study. Nevertheless, we evaluated the cases using a longitudinal analysis and with multivariate analysis.

## 5. Conclusions

Several of the evaluated proteins including CD44v6, p53, p73, and podoplanin showed an increasing expression from normal tissue to KUS and HG cases, suggesting a possible role in oral cancer development. Podoplanin was independently associated with the malignant transformation of OL and using alone or combined with the binary dysplasia grade and could be a useful predictive biomarker of malignant transformation in OL.

## Figures and Tables

**Figure 1 biomolecules-12-00606-f001:**
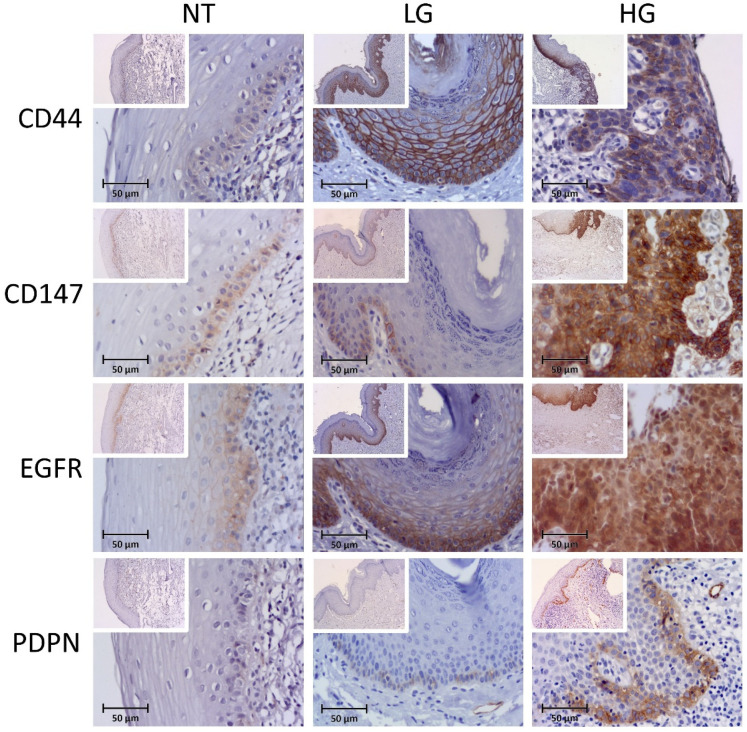
Immunohistochemical staining of the CD44v6, CD147, EGFR, podoplanin (PDPN) proteins in normal tissues (NT); low-grade (LG), and high-grade (HG) cases. Small boxes inside the main photo (magnification ×200) correspond to the original image at magnification ×10. High expression can be observed for all cases of HG and for CD44v6 and EGFR for LG cases.

**Figure 2 biomolecules-12-00606-f002:**
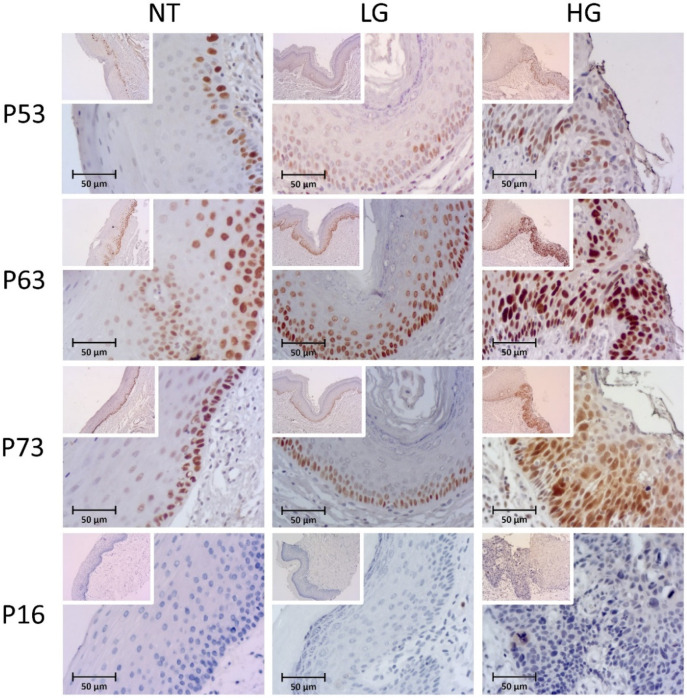
Immunohistochemical staining of the P53, P63, P73, and P16 proteins in normal tissues (NT), low-grade (LG), and high-grade (HG) cases. Small boxes inside the main photo (magnification ×200) correspond to the original image at magnification ×10. High expression can be observed for P53, P63, and P73 in HG, and for p63 in NT and LG cases.

**Figure 3 biomolecules-12-00606-f003:**
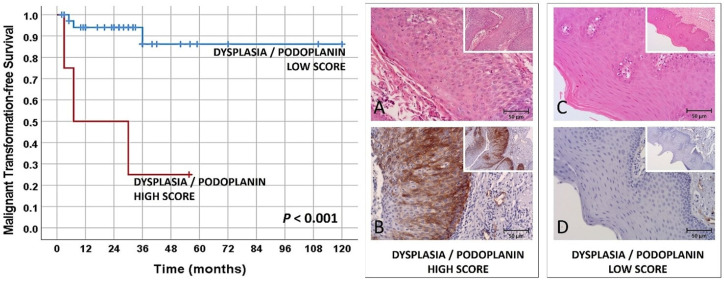
Kaplan–Meier analysis of oral malignant transformation-free survival in patients with oral leukoplakia using a combined score of binary dysplasia grade and podoplanin expression (*p* < 0.001 Kaplan–Meier analysis). (**A**,**B**) Oral leukoplakia of high dysplasia grade and high expression of podoplanin; (**C**,**D**) oral leukoplakia with low dysplasia grade and low expression of podoplanin; (**A**,**C**) hematoxylin-eosin staining; and (**B**,**D**) podoplanin immunostaining. Small boxes inside the main photos (magnification ×200) correspond to the original image at magnification ×100.

**Table 1 biomolecules-12-00606-t001:** Primary monoclonal antibodies used for immunohistochemistry in the study.

Primary Antibody	Clone	Dilution	Manufacture
anti-CD44v6	VFF-7	1:120	Leica Biosystems, Newcastle upon Tyne, UK
anti-CD147	AB1843	1:30	Leica Biosystems, Newcastle upon Tyne, UK
anti-EGFR	EGFR.25	1/100	Leica Biosystems, Newcastle upon Tyne, UK
anti-p53	NCLp53 DO7	1:20	Leica Biosystems, Newcastle upon Tyne, UK
anti-p63	7JUL	1:25	Leica Biosystems, Newcastle upon Tyne, UK
anti-p73	24	1:25	Leica Biosystems, Newcastle upon Tyne, UK
anti-p16	OA315	prediluted	MTM, Heidelberg, Germany
anti-podoplanin	D2-40	1:150	Dako, Carpinteria, CA, USA

**Table 2 biomolecules-12-00606-t002:** Univariable analysis of the influence of variables in the oral malignant transformation-free survival using Kaplan–Meier curves.

Variable		N	Oral Malignant Transformation	MTFS ^a^	*p*-Value ^b^	*p*-Value ^c^
CD44	0, I II, III	2 45	0 6	100 80.1	0.510	0.803
CD147	0, I II, III	20 24	1 3	94.7 83.3	0.586	0.779
EGFR	0, I II, III	2 43	0 5	100 82.7	0.610	0.461
P53	0, I II, III	14 25	1 3	92.9 82.1	0.574	0.642
P73	0, I II, III	11 30	2 3	75.8 37.5	0.582	0.526
P63	0, I II, III	1 42	0 5	100 85.2	0.745	0.478
P16	0, I II, III	32 8	3 3	88.4 58.3	0.081	0.126
P16 TMN	0–74 >75%	38 2	6 0	78.3 100	0.550	-
Podoplanin	0, I II, III	34 5	2 4	88.7 20	<0.001	<0.001
Podoplanin ^d^	0, I II, III	22 16	1 5	85.7 65.5	0.044	0.058

MTFS, Malignant Transformation-free Survival; ^a^ Percentage of cases without event at 5 years of follow-up (Kaplan–Meier estimates of probability of survival); ^b^ Log-rank test for biomarkers using I+II vs. III+IV cut-off score; ^c^ Log-rank test for biomarkers using four categories’ score without cut-off. ^d^ using Kawaguchi et al. [31] score.

## Data Availability

The data presented in this study are available on request from the corresponding author.

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
