# Peer review of "Podoplanin Expression Independently and Jointly with Oral Epithelial Dysplasia Grade Acts as a Potential Biomarker of Malignant Transformation in Oral Leukoplakia"

_biomolecules, 2022, doi:10.3390/biom12050606_

Round 1
Reviewer 1 Report
Luís Monteiro and colleagues evaluate the expression of several biomarkers in oral leukoplakias (OL), including 41 low-grade and 11 high-grade cases. They found podoplanin expression combined with dysplasia grade could be useful predictive markers of malignant transformation in OL. However, the sample size is relatively small, and a previous study (Kawaguchi et al) has reported an association of podoplanin with MT in OL with a larger sample size (150 OL patients). In addition, there are several points the authors need to clarify.
Major:
- Tissue dysplasia was divided into low-grade and high-risk grade OL based on the binary histological grade classification. The authors should provide more details about the binary histological grade classification.
- Authors evaluate the expression of several biomarkers, but the rationale of how these biomarkers were chosen was missing.
- The fact that several markers were not detected in some cases worries me a lot. Does it mean the expression level is too low to be detected? Or is it caused by the technique variabilities?
- Authors quantify the high expression of each marker in NT, LG, and HG cases. It remains not clear how the high expression is defined. It is not clear about the threshold. Figures include the representative image from each group. It would be helpful if the authors showed a collection of different expression levels and define the one with high expression, which would be quantified and analyzed.
Minor:
- Please shorten the title.
Author Response
Our responses to the reviewers and revisions of the manuscript.
Response to Reviewer 1 Comments
Point 1:
Luís Monteiro and colleagues evaluate the expression of several biomarkers in oral leukoplakias (OL), including 41 low-grade and 11 high-grade cases. They found podoplanin expression combined with dysplasia grade could be useful predictive markers of malignant transformation in OL. However, the sample size is relatively small, and a previous study (Kawaguchi et al) has reported an association of podoplanin with MT in OL with a larger sample size (150 OL patients). In addition, there are several points the authors need to clarify.
Response 1: Dear reviewer, thank you for your comments. In 2022, several systematic reviews have point out the lack of observational studies to reveal the scientific evidence of biomarkers in OL prognostification. Podoplanin, although reported by Kawaguchi et al, 2008, has been pointed out as one of possible biomarkers related with MT in OL their work has not been reproduced in other settings. Morever, we present a combination of biomarkers such as p53, p63, EGFR, CD44 and, showing the value of podoplanin (when compared with other biomarkers) in the prognostification of MT in this sample of OL. Nevertheless, and taking the comments of the reviewer, we mention the small sample size as a liimitation in our discussion.
Point 2:
Tissue dysplasia was divided into low-grade and high-risk grade OL based on the binary histological grade classification. The authors should provide more details about the binary histological grade classification.
Response 2: we include now the caractheristics of the binary histological grade classification used. We revised the text by adding::
“In this classification the cut-off point between “high-risk’’ vs “low-risk” lesions was made considering at least four architectural changes and five cytological changes for “high-risk” lesions (including the architecture criteria: 1 - irregular epithelial stratification; 2 - loss of polarity of basal cells; 3 - drop-shaped rete ridges; 4 - increased number of mitotic figures; 5 - abnormally superficial mitoses; 6 - premature keratinisation in single cells; 7 - keratin pearls within rete ridges; and cytology criteria: 1 - abnormal variation in nuclear size; 2 - abnormal variation in nuclear shape; 3 - abnormal variation in cell size; 4 - abnormal variation in cell shape; 5 - increased nuclear– cytoplasmic ratio; 6 - increased nuclear size; 7 - atypical mitotic figures; 8 - increased number and size of nucleoli; 9 – hyperchromatism).”
Point 3:
Authors evaluate the expression of several biomarkers, but the rationale of how these biomarkers were chosen was missing.
Response 3: We include in the introduction the reasons to explain why we choose these biomarkers as follows:
“. Many of these deregulated genes and proteins could be indicators of an active ongoing carcinogenesis process and could act as potential predictive biomarkers in OPMD [17, 19-21]. We have previously reported the abnormal expression of several proteins in oral squamous cell carcinomas, such as EGFR, p53, p16, podoplanin or CD147 [22-24], and we hypothesized “could these proteins be abnormally expressed in OL?”…… “and more research should address the role of these biomarkers in prognostification of OPMD´s.”
Point 4:
The fact that several markers were not detected in some cases worries me a lot. Does it mean the expression level is too low to be detected? Or is it caused by the technique variabilities?
Response 4: We acknowledge that in some cases the biomarkers were not detected or expression as low, but this is a result we accept as the controls used showed positive results. We excluded invalid cases, without epithelial tissue. The others that, had epithelial tissue were evaluated and were valid we evaluated appropriately. We provide the maximum cases per run to avoid technique differences and always using positive and negative controls in each run of experiments to control this phenomenon. However, this is a well-known potential limitation of using IHC as previously reported. We include this aspect in limitations part of discussion.
Point 5:
Authors quantify the high expression of each marker in NT, LG, and HG cases. It remains not clear how the high expression is defined. It is not clear about the threshold. Figures include the representative image from each group. It would be helpful if the authors showed a collection of different expression levels and define the one with high expression, which would be quantified and analyzed.
Response 5: we include now a better description of the cut-offs used and of illustrations specially for figure 3. We already included several of the images regarding the evaluated biomarkers showing high vs low expression for all biomarkers in figure 1, figure 2 and figure 3. We now provide a more detailed description of these images.
Point 6:
Please shorten the title.
Response 6: We understand that this is a long title but as it is, it gives the correct idea of the scope of our study, so we prefer to maintaint like this, if possible. Thank you for your comprehension on this.
Thank you for all your comments.

Reviewer 2 Report
Dear authors, this study is well designed and provide an elaborative explanation on the expression of the biomarkers in oral mucosa tissues with premalignant lesions as possible markers for oral cancer. The methodology is sound and the results adequately presented. I suggest the addition of the following paper in the discussion (https://doi.org/10.1111/odi.13747) as it is rather recent and provide a better understanding on the biomarkers that could be potentially used in the clinical practice.
Author Response
REVIEWER 2
Point 1:
Dear authors, this study is well designed and provide an elaborative explanation on the expression of the biomarkers in oral mucosa tissues with premalignant lesions as possible markers for oral cancer. The methodology is sound and the results adequately presented. I suggest the addition of the following paper in the discussion (https://doi.org/10.1111/odi.13747) as it is rather recent and provide a better understanding on the biomarkers that could be potentially used in the clinical practice.
Response 1: Dear reviewer thank you for your kind words on our article. We have now included the suggested article (ref no 21).

Reviewer 3 Report
The present article speaks about the expression of biomarkers, CD44v6, CD147, EGFR, p53, p63, 19p73, p16 and podoplanin in oral leukoplakias (OL) and tried to assess their potential for prediction of malignant transformation (MT).
This is a very interesting article well written with relevant significance for the literature.
I would just suggest the authors to add more recent published articles in the discussion section.
Author Response
Reviewer 3
Point 1:
The present article speaks about the expression of biomarkers, CD44v6, CD147, EGFR, p53, p63, 19p73, p16 and podoplanin in oral leukoplakias (OL) and tried to assess their potential for prediction of malignant transformation (MT). This is a very interesting article well written with relevant significance for the literature. I would just suggest the authors to add more recent published articles in the discussion section.
Response 1: Dear reviewer thank you for your kind words on our article. We have now cited more recent articles in the discussion (ref nos 21, 58, 68,69, 70 and 75)..

Round 2
Reviewer 1 Report
The authors are responsive and have clarified most of the previous critiques.